# Rapid Antibacterial Activity Assessment of Chimeric Lysins

**DOI:** 10.3390/ijms25042430

**Published:** 2024-02-19

**Authors:** Jin-Mi Park, Jun-Hyun Kim, Gun Kim, Hun-Ju Sim, Sun-Min Ahn, Kang-Seuk Choi, Hyuk-Joon Kwon

**Affiliations:** 1Laboratory of Poultry Medicine, College of Veterinary Medicine, Seoul National University, Seoul 08826, Republic of Korea; niji17@snu.ac.kr (J.-M.P.); loupgarouhs@snu.ac.kr (J.-H.K.); vicky.ahn@snu.ac.kr (S.-M.A.); 2College of Veterinary Medicine and BK21 for Veterinary Science, Seoul National University, Seoul 08826, Republic of Korea; smilesssss@snu.ac.kr (G.K.); gjswn0302@snu.ac.kr (H.-J.S.); 3GeNiner Inc., Seoul 08826, Republic of Korea; 4Laboratory of Pharmacology, Research Institute for Veterinary Science, College of Veterinary Medicine, Seoul National University, Seoul 08826, Republic of Korea; 5Laboratory of Avian Diseases, College of Veterinary Medicine, Seoul National University, Seoul 08826, Republic of Korea

**Keywords:** *Staphylococcus aureus*, chimeric lysins, cell-free expression system, linker optimization, SH3 shuffling, antibacterial activity ranking

## Abstract

Various chimeric lysins have been developed as efficacious antibiotics against multidrug-resistant bacteria, but direct comparisons of their antibacterial activities have been difficult due to the preparation of multiple recombinant chimeric lysins. Previously, we reported an *Escherichia coli* cell-free expression method to better screen chimeric lysins against *Staphylococcus aureus*, but we still needed to increase the amounts of expressed proteins enough to be able to detect them non-isotopically for quantity comparisons. In this study, we improved the previous cell-free expression system by adding a previously reported artificial T7 terminator and reversing the different nucleotides between the T7 promoter and start codon to those of the T7 phage. The new method increased the expressed amount of chimeric lysins enough for us to detect them using Western blotting. Therefore, the qualitative comparison of activity between different chimeric lysins has become possible via the adjustment of the number of variables between samples without protein purification. We applied this method to select more active chimeric lysins derived from our previously reported chimeric lysin (ALS2). Finally, we compared the antibacterial activities of our selected chimeric lysins with reported chimeric lysins (ClyC and ClyO) and lysostaphin and determined the rank orders of antibacterial activities on different *Staphylococcus aureus* strains in our experimental conditions.

## 1. Introduction

*Staphylococcus aureus* (*S. aureus*) is a Gram-positive bacteria and a causative agent of bovine mastitis, arthritis in chickens, and food poisoning in humans, causing economic losses and public health problems [1,2,3]. The worldwide economic losses caused by *S. aureus* are estimated to be from USD 19.7 billion to USD 32 billion per year [4]. Multidrug resistance is steadily increasing worldwide, and the death toll of methicillin-resistant *S. aureus* (MRSA) infections in humans was 120,000 in 2019 [5,6,7,8,9]. Humans are regarded as the natural host for *S. aureus* due to the presence of more diverse genotypes than in animals; however, spill-over strains have adapted to animals by acquiring mutations and horizontally transferred genetic elements [10,11]. Zoonotic and reverse zoonotic cases of multidrug-resistant and pathogenic *S. aureus* have been reported, and alternatives to conventional antibiotics are desperately needed [2,12,13]. Bacteriophages are emerging as a powerful alternative to conventional antibiotics due to their specific and prolonged amplifying antibacterial activity [14,15,16]. However, various resistance mechanisms against bacteriophages (phages) have been developed by host bacteria, some of which are horizontally transferred [17].

Since the discovery of phage endolysin as an antibacterial agent in 2001, various natural and chimeric endolysins have been developed as recombinant proteins for alternative antibiotics [18,19,20]. Natural endolysins are composed of two or more modular domains with catalytic activities to cleave the linkages between peptidoglycan (PEG) multilayers and binding activity to PEG in the cell wall [16,18,21]. Most staphylococcal phage endolysins have three domains: cysteine- and histidine-dependent amidohydrolase/peptidase (CHAP), a central N-acetylmuramyl-L-alanine amidase (Ami_2 or Ami_3), and SH3b (cell wall binding domain, CBD) [16,18,21,22]. Their modular structure allows recombination with autolysins or other phage lysins to generate new chimeric lysins. The development of chimeric lysins offers the possibility of finding new enzymes with enhanced functions, such as an improved soluble protein expression, increased lytic activity, and improved host range [22,23,24,25]. Therefore, intensive efforts to find better chimeric lysins have been conducted via random domain shuffling and antibacterial activity tests using *Escherichia coli* (*E. coli*)-expressed proteins or via induced lysis-based screening [25,26,27,28,29]. Gene cloning into plasmid vectors and *E. coli* expression are laborious processes, and simpler and more rapid in vitro screening and activity-comparing protocols have been developed [30,31]. Previously, we developed a chimeric lysin (ALS2), which is composed of the CHAP domain of an autolysin (SsaA) and the amidase and SH3 domains of *ϕ*K lysin (LysK), after screening multiple candidates using a simple and rapid cell-free expression system [31]. However, the cell-free expression system needs to be improved for higher-expression-level proteins.

In this study, we improved our cell-free expression system by adding a previously reported, strong, artificial T7 terminator and reverting various mutations between the T7 promoter and start codon to wild-type nucleotides of *ϕ*T7 [32]. The improved cell-free expression system increased the expression levels of chimeric lysins and allowed us to compare the expressed amounts of proteins using Western blotting and a densitometer. Therefore, we could directly compare the antibacterial activities of different chimeric lysins by adjusting their concentrations the same. We could improve the antibacterial activity of ALS2 by deleting the amidase domain and optimizing the linker between the CHAP and SH3 domains, and shuffling with various SH3 domains [23,27,28]. Additionally, we directly compared the antibacterial activities of reported chimeric lysins (ClyC and ClyO) and lysostaphin and determined their rank orders [24,33,34].

## 2. Results and Discussion 

### 2.1. Improvement of the Cell-Free Expression System by Adding an Artificial T7 Terminator

The 10 genes encoding the major capsid of the T7 phage are distinguished by the presence of the downstream T7 terminator (T*ϕ*) for efficient transcription termination [35]. *E. coli* expression vectors and the cell-free expression system have adopted the T7 terminator to increase the expression of the target genes, but the termination rate of T*ϕ* is incomplete (62%), and an artificial T*ϕ* was recently reported to be more efficient (91%) than T*ϕ* [32]. We adapted the artificial T*ϕ* to our cell expression system to improve the transcription efficiency. Additionally, we converted all the artificial nucleotide sequences around the T7 promoter to natural ones and added the six-histidine tag to detect the produced chimeric lysins via Western blotting. The schematic gene structure of our cell-free expression system is shown in Figure 1A. The gene template produced via SOE-PCR was used for *E. coli* cell-free expression. To compare the differences in gene expression, we prepared a chimeric lysin gene (ALS2-dA-L31), which we generated by deleting the amidase domain and seven amino acids of the linker of ALS2 (Figure 1F). The gene templates of ALS2-dA-L31, which were prepared using both the present (P_ALS2-L31+T term) and previous (V_ALS2-L31+TAG) methods, were expressed, and the antibacterial activities were compared using a turbidity reduction test [31]. In contrast to V_ALS2-dA-L31+TAG and the negative control, the P_ALS2-dA-L31+T term showed significantly lower OD_600_ values, reflecting higher antibacterial activity against the different genotypes of human MRSA strains: CCARM3806 (RST10-2) and CCARM3840 (RST4-1) (Figure 1C) [31]. Although we did not test the individual effects of the T7 terminator and the modified nucleotide sequences, they may increase the expression of ALS2-dA-L31 via the optimization of the transcription efficiency of T7 polymerase. 

### 2.2. Comparison of Antibacterial Activities of ALS2-dA Variants with Different Linker Lengths

The hypothetical structure and amino acid sequence of the linker between the amidase and SH3 domains of ALS2 is shown in Figure 1E. The ten variants of ALS2-dA were generated, with different linkers in terms of length and amino acid sequence (Figure 1F). The number of DNA templates and expressed proteins are compared in Figure 2. Although similar amounts of the ten DNA templates were used for cell-free expression, the amounts of expressed proteins were different from each other by up to 5-fold (Figure 2A–C). The relative amounts of the expressed proteins were represented with relative intensities (RIs), which were calculated on the basis of the band intensity of ALS2-dA-L21, using a densitometer (Figure 2C). The ALS2-dA-L25 (RI = 0.95) and ALS2-dA-L26 (RI = 0.93) were expressed similarly to ALS2-dA-L21 (RI = 1.00), but others, especially ALS2-dA-L34 (RI = 0.23), ALS2-dA-L38 (RI = 0.33), and ALS2-dA-L24 (RI = 0.35), showed very low expression levels.

The antibacterial activities of the ten linker variants were compared via turbidity reduction tests against four different human MRSA strains: CCARM3806 (RST10-2), CCARM3826 (RST2-1), CCARM3832 (RST2-1), and CCARM3837 (RST4-1), without adjusting the amounts of proteins (Figure 2D). We repeated the tests with volumes that differed by 2-fold (2, 1, and 0.5), with all the chimeric lysins at the same volume for each test. ALS2-dA-L25 and ALS2-dA-L35 showed significantly lower OD_600_ values than the negative controls for all the tested strains. Considering similar amounts of proteins, the antibacterial activities of ALS2-dA-L21 and ALS2-dA-L26 are apparently lower than that of ALS2-dA-L25. Although we did not compare the antibacterial activities between ALS2-dA-L25 and ALS2-dA-L35 by adjusting the amounts of proteins, ALS2-dA-L35 did not perform better than ALS2-dA-L25 in the repeated antibacterial activity testing. Therefore, we selected ALS2-dA-L25 for further study due to its better cell-free expression. The role of linkers in proteins is known to affect the activity [36], stability [37], and even folding [38] of proteins, in addition to the connection of domains [37,39]. As shown in Figure 2, the length of the linker has been shown to have apparent effects on chimeric lysin expression and antibacterial activity. The selection of suitable linkers to join protein domains can be complex and is often neglected in the design of fusion proteins. Our screening method can serve as a useful tool to find the optimal linker by optimizing the length of natural linkers. High and even expressions of chimeric lysins in the cell-free expression condition are important for downstream tests, but the dramatic changes in expression of approximately 2- to 5-fold due to the different linkers in the middle of the gene were unexpected, as the GC contents and sequences of mRNA bases +3 to +25 are currently seen as important components of the translation machinery for binding and affecting expression levels in genes [40].

### 2.3. Antibacterial Activity of E. coli-Expressed ALS2-dA-L25 against Bovine, Poultry, and Human Strains of S. aureus

ALS2-dA-L25 was expressed in *E. coli* and purified using a Ni-NTA column. The antibacterial spectrum of ALS2-dA-L25 was tested via plate lysis and turbidity reduction tests with 20 previously reported *S. aureus* strains from bovine mastitis, chicken arthritis, and human infection (Table 1) [41,42,43].

We graded the antibacterial activities as strong (+++), intermediate (++), weak (+), and negative (-) (Figure 3A), and 10 μg of ALS2-dA-L25 formed clear zones from weak to strong in all the tested strains. All the bovine mastitis strains (PMB series) and chicken arthritis strains (SNU series), and six out of ten human infection strains (CCARM3105, CCARM3806, CCARM3837, CCARM3840, CCARM3865, and CCARM3897) were relatively susceptible to showing strong lysis zones at 10 μg. In particular, PMB2-1 and SNU19045 and CCARM3837 were highly susceptible to showing strong or intermediate lysis zones even at 0.5 μg. In contrast to the others, CCARM3805, CCARM3825, and CCARM3832 showed weak lysis even at 10 μg. In turbidity reduction tests, the antibacterial activity was represented with turbidity ratios (TRs), which constitute the OD_600_ value ratio of ALS2-dA-L25-treated to untreated samples. All the tested strains had variable antibacterial activities, showing TRs between 0.17 and 0.92 (less than 1). Except for CCARM3897, all the strains that showed a strong lysis zone at 10 μg showed TRs of less than 0.5. CCARM3805, CCARM3825, and CCARM3832 showed higher TRs than the others. Therefore, the different susceptibilities of different strains of *S. aureus* were again confirmed, and human RST2-1 strains (3/4) were shown to be resistant to ALS2-dA-L25 [31].

In our previous study, ALS2 did not show a clear lysis zone in PMB66-1 and PMB67-1 even at 10 ug, but ALS2-dA-L25 showed clear lysis zones. Additionally, ALS2 showed high TRs (>0.8) in PMB66-1, PMB67-1, PMB179-1, PMB242-1, CCARM3837, and CCARM3840, but ALS2-dA-L25 showed lower TRs of between 0.19 and 0.49. Therefore, the removal of the amidase domain and a shortened linker length may increase the antibacterial activity. These results are consistent with other studies [27,28].

### 2.4. Improvement of Antibacterial Activity of ALS2-dA-L25 by Shuffling SH3 Domains

The combination of the appropriate SH3 domain with the catalytic domain is important for the development of efficacious chimeric lysins [16,24,45]. We selected seven SH3 domains, of which six (ClyC_SH3_, vB_SH3_, tras14_SH3_, ClyO_SH3_, 2638A_SH3_, and 2958_SH3_) originated from phage lysins, and one (Lsp_SH3_) originated from lysostaphin (Appendix A). The amino acid sequences of the selected SH3 domains were aligned and compared with the SH3 domain (K_SH3_) of ALS2-dA-L25 (Figure 4A). The amino acid lengths of the compared SH3 domains ranged from 80 to 87 amino acids. The amino acid sequences of ClyC_SH3_ and vB_SH3_ were 94.2% and 93.1%, identical to K_SH3_, respectively. The other five SH3 domains showed very low similarity to K_SH3_, ranging from 40.2% to 4.5%. Each SH3 domain replaced the K_SH3_ of ALS2-dA-L25 and was linked directly to the CHAP domain and the linker, L25, of ALS2-dA-L25.

The antibacterial activities of new SH3 chimeras (ALS2-dA-ClyC_SH3_, ALS2-dA-vB_SH3_, ALS2-dA-tras14_SH3_, ALS2-dA-ClyO_SH3_, ALS2-dA-Lsp_SH3_, ALS2-dA-2638A_SH3_, and ALS2-dA-2958_SH3_) were compared with ALS2-dA-L25 using a turbidity reduction test with the four human MRSA strains: CCARM3806 (RST10-2), CCARM3825 (RST2-1), CCARM3832 (RST2-1), and CCARM3837 (RST4-1), which showed different resistances to ALS2-dA-L25 (Figure 4B). To compare the relative antibacterial activity, the relative amounts of chimeric lysins were represented by relative intensities (RIs) to the highly expressed ALS2-dA-2638A_SH3_ (RI = 1.0) and were measured via Western blotting and a densitometer (Figure 4C,D). Other than ALS2-dA-2958_SH3_ (RI = 0.09), ALS2-dA-L25 showed the lowest RI (RI = 0.32) and was similar to those of ALS2-dA-ClyC_SH3_ (RI = 0.40) and ALS2-dA-vB_SH3_ (RI = 0.36), which have similar amino acid sequences. The relative amounts of ALS2-dA-tras14_SH3_ (RI = 0.57), ALS2-dA-ClyO_SH3_ (RI = 0.64), ALS2-dA-Lsp_SH3_ (RI = 0.91), and ALS2-dA-2638A_SH3_ (RI = 1.00) were higher than ALS2-dA-ClyC_SH3_. The lack of antibacterial activity of ALS2-dA-2958_SH3_ may be due to the extremely low amount of protein. Considering the highest amount of ALS2-dA-2638A_SH3_ with no antibacterial activity, the 2638A_SH3_ is not compatible with the CHAP and L25 of ALS2-dA-L25. However, other chimeric lysins showed significantly high antibacterial activities against at least one out of the tested strains. The antibacterial activity of ALS2-dA-ClyC_SH3_ was best in three human MRSA strains: CCARM3806 (RST10-2), CCARM3832 (RST2-1), and CCARM3837 (RST4-1), and we selected ALS2-dA-ClyC_SH3_ for further study in this paper, as the previous report’s optimal SH3 domain combination increased the antibacterial activity of chimeric lysins [16,24,45]. In addition, it is noteworthy that different SH3 domains with different amino acid sequences may affect the expression levels of genes in the cell-free expression condition.

### 2.5. Comparison of Antibacterial Activities of E. coli-Expressed ALS2, ALS2-dA-L25, and ALS2-dA-ClyC_SH3_

ALS2, ALS2-dA-L25, and ALS2-dA-ClyCSH3 were expressed in *E. coli* and purified using Ni-NTA columns to determine their minimal inhibitory concentrations (MICs) against representative strains from different hosts (Table 2). The final protein concentrations of ALS2, ALS2-dA-L25, and ALS2-dA-ClyC_SH3_ were 1.1 mg/mL, 2.24 mg/mL, and 1.91 mg/mL, respectively, but they contained other minor protein bands (Appendix A). We measured the densities of all bands and calculated the purity of each chimeric lysin to adjust the amounts of chimeric lysins. The purities of ALS2, ALS2-dA-L25, and ALS2-dA-ClyC_SH3_ were 72.4%, 91.1%, and 80.2%, respectively (Table 2). Based on the purity-adjusted concentrations of chimeric lysins, ALS2-dA-L25 (2.0 mg/mL) was 1.3-fold and 2.5-fold more than ALS2-dA-ClyC_SH3_ (1.5 mg/mL) and ALS2 (0.8 mg/mL), respectively. The purity-adjusted MICs against PMB4-1, SNU19045, and CCARM3806 are summarized in Table 2. ALS2-dA-L25 had approximately 4.8-, 3.2-, and 2.4-fold lower MICs than ALS2 for the tested strains. ALS2-dA-ClyC_SH3_ had 1.9-, 2.2-, and 1.5-fold less MICs than ALS2-dA-L25. The Western blotting results showed the presence of sub-proteins of ALS2, which contained an SH3-containing C-terminal part (Appendix A). Although the sub-proteins have no cell wall lysis activities, they may decrease the antibacterial activity of ALS2 by competing for binding sites. Due to their minor proportion, their reducing effects on the antibacterial activity of ALS2 were recorded as negligible so as not to mislead from the conclusion that the antibacterial activity of ALS2 is lower than those of ALS2-dA-L25 and ALS2-dA-ClyC_SH3_. For all chimeric lysins, the susceptibilities of strains isolated from bovine mastitis (PMB4-1), poultry arthritis (SNU19045), and human infection (CCARM3806) decreased in that order. The antibacterial activity and sub-protein problem of parental ALS2 were demonstrated to be improved via amidase deletion and linker optimization, and the replacement of K_SH3_ with ClyC_SH3_ was confirmed to improve antibacterial activity [24,27,28,31,45]. However, the antibacterial activity of the commercial recombinant lysostaphin was 18-, 38-, and 100-fold stronger than that of ALS2-dA-ClyC_SH3_ for the tested strains.

### 2.6. Simple and Rapid Direct Comparison of Antibacterial Activities of Chimeric Lysins Using the Improved Cell-Free Expression System

To date, the antibacterial activities of various reported chimeric lysins have been compared with lysostaphin, but we alone have directly compared their relative activities. To encourage the development of better chimeric lysins, simple and rapid direct comparisons of antibacterial activities with reported chimeric lysins are essential. For this reason, we applied the improved cell-free expression system to compare the antibacterial activities of chimeric lysins. We prepared DNA templates of the recently reported chimeric lysins, ClyC and ClyO, and compared their antibacterial activities with those of ALS2-dA-L25, ALS2-dA-ClyC_SH3_, and lysostaphin. Additionally, we generated a DNA template of the new chimeric lysin Lsp-ClyC_SH3_, composed of the peptidase and the linker of lysostaphin, and the SH3 domain of ClyC, for comparison (Appendix A).

At first, we worried about the stability of the chimeric lysins, so we tested their antibacterial activities using a turbidity reduction test just after a cell-free expression reaction with 2-fold dilutions (Figure 5A). Then, we performed Western blotting and measured the density of each band to calculate the RI of each chimeric lysin in reference to that of lysostaphin (RI = 1.00) (Figure 5). The RIs of ALS2-dA-L25, ALS2-dA-ClyC_SH3_, ClyC, ClyO, and Lsp-ClyC_SH3_ were 0.29, 0.54, 0.17, 2.08, and 0.10, respectively. The relative antibacterial activity of each chimeric lysin was represented by the adjusted minimal inhibitory volumes, which were calculated by multiplying the completely inhibited volumes (MIVs) and the RIs of the chimeric lysins (Table 3). The experimental MIVs for CCARM3806 increased ALS2-dA-ClyC_SH3_/lysostaphin/ClyC (0.25), ALS2-dA-L25/Lsp-ClyC_SH3_ (0.5), and ClyO (1.0), in that order, but the adjusted MIVs increased ClyC (0.04), Lsp-ClyC_SH3_ (0.05), ALS2-dA-ClyC_SH3_ (0.14), ALS2-dA-L25 (0.15), lysostaphin (0.25), and ClyO (2.08), in that order. The adjusted MIVs for CCARM3825 increased Lsp-ClyC_SH3_ (0.05), lysostaphin (0.25), ALS2-dA-ClyC_SH3_ (0.54), and ClyO (1.04), in that order, but the values for ALS2-dA-L25 (>0.29) and ClyC (>0.17) could not be determined. The adjusted MIVs for CCARM3832 increased Lsp-ClyC_SH3_ (0.05), ClyC (0.09), lysostaphin (0.25), and ALS2-dA-ClyC_SH3_ (0.54), in that order, but the values for ALS2-dA-L25 (>0.29) and ClyO (>2.08) could not be determined. The adjusted MIVs for CCARM3837 increased ClyC (0.04), Lsp-ClyC_SH3_ (0.05), lysostaphin (0.25), ALS2-dA-ClyC_SH3_ (0.27), ALS2-dA-L25 (1.00), and ClyO (2.08), in that order. The antibacterial activities of ClyC and Lsp-ClyC_SH3_ were similar for CCARM3806 and CCARM3837, but those of Lsp-ClyC_SH3_ were 1.8-fold and more than 3.4-fold higher than ClyC for CCARM3825 and CCARM3832. In contrast to the highest antibacterial activity of commercial recombinant lysostaphin used in Table 2, the lysostaphin expressed in the cell-free system showed much lower antibacterial activity than ALS2-dA-ClyC_SH3_ for CCARM3806.

After confirming the stability of the tested cell-free-expressed chimeric lysins for several days at 4 °C, we first determined the RIs of the chimeric lysins, then adjusted the concentrations of chimeric lysins to be the same as the lowest, that of Lsp-ClyC_SH3_. The cell-free expression reaction contains nutrients, including 20 amino acids and NTPs, and facilitates bacterial growth. To minimize the effects of different amounts of remaining nutrients in chimeric lysins, we diluted high-concentration chimeric lysins (exhausted nutrients) via a DNA-negative, cell-free expression reaction to yield the same concentration of Lsp-ClyC_SH3_ and used the same volumes of chimeric lysins. We compared their antibacterial activities in 2-fold dilutions using turbidity reduction tests. Lsp-ClyC_SH3_ inhibited the growth of all the tested strains at 0.5 μL, but ClyC and lysostaphin did so for only two strains (CCARM3806 and CCARM3837) and one strain (CCARM3837), respectively. Therefore, Lsp-ClyC_SH3_ was verified as having stronger antibacterial activities and a broader antibacterial spectrum than ClyC and lysostaphin for CCARM3825 and CCARM3832 (Figure 6). Thus, the antibacterial activity of Lsp-ClyC_SH3_ was the best among the tested chimeric lysins.

Interestingly, the antibacterial activities of ALS2-dA-L25 and ALS2-dA-ClyC_SH3_ against CCARM3806 were comparable to that of lysostaphin (Figure 6), and the result was unexpected due to the superior antibacterial activity of commercial recombinant lysostaphin to *E. coli*-expressed ALS2-dA-L25 and ALS2-dA-ClyC_SH3_ (Table 2). The difference can be partially explained by the differences in purity between commercially purified lysostaphin (approximately 91%) and chimeric lysins, but there may be another major reason for the result. The recombinant lysostaphin has no His-tag and was purified using cation-exchange columns [34]. We confirmed that the commercial recombinant lysostaphin was not detected by the anti-His-tag antibody in Western blotting. To date, most chimeric lysins have been expressed with a His-tag, and the His-tag is known to be neutral to the antibacterial activity [24,46]. On the contrary, the added His-tag for the detection of chimeric lysins may decrease the antibacterial activity. Therefore, the effect of the His-tag on the antibacterial activity of chimeric lysins needs to be unraveled in order to improve the activity of chimeric lysins. Although the conditions for antibacterial activity comparison may not have been optimal for all the tested chimeric lysins, our first attempt to compare the antibacterial activities of newly reported or licensed chimeric lysins under the same conditions may be important for encouraging the development of better chimeric lysins to overcome multidrug-resistant bacteria.

### 2.7. Comparison of Antimicrobial Activities of Chimeric Lysins in Milk

Bovine mastitis caused by *S. aureus* is important due to issues around economic loss and food safety in the dairy industry. A reverse zoonotic human genotype of the *S. aureus* strain has been reported in bovine mastitis, and the MRSA strains tested in this study could also be potential pathogens in cows. Therefore, the antibacterial activities of the chimeric lysins in MRSA-spiked milk were tested for comparison (Figure 7). Milk is a colloid in which colloidal entities, such as fat globules, casein micelles, whey proteins, and lactose, ranging from 1 to 2000 nm in diameter, are dispersed in water, and the antibacterial activities of chimeric lysins may be different from those in TSI broth [47]. The TSI broth-based turbidity reduction test is sensitive to the added amount of cell-free expression reaction due to the change in nutrients for the inoculated bacteria, but milk is less sensitive to the added cell-free reaction. Although the final concentration of each chimeric lysin was slightly different due to the changes in final volumes (e.g., lysostaphin: 101 μL and ALS2-dA-ClyC_SH3_: 110 μL), the different volumes of chimeric lysins were added according to the RIs of chimeric lysins to adjust the different amounts of chimeric lysins (Figure 5B). According to the colony reduction test, lysostaphin and Lsp-ClyC_SH3_ showed higher antibacterial activities against all the tested strains in milk than the others did. ClyC showed good antibacterial activity against all three strains, except CCARM3825, in accordance with the turbidity reduction test (Figure 5A and Figure 6C). ALS2-dA-L25 and ALS2-dA-ClyC_SH3_ showed high antibacterial activity against CCARM3806 and CCARM3837, but showed no colony reduction against CCARM3825 and CCARM3832, which are RST2-1 genotypes. The RST2-1 genotypes are prevalent in humans, and the CHAP domain that originated from a commonly shared autolysin, SsaA, via *S. aureus* strains may be out of date in terms of overcoming the resistance mechanisms acquired by the prevalent strains [31]. ClyO did not show any significant antibacterial activities in any of the tested strains. Although lower numbers of bacteria (10^2^ CFU/100 μL/well) were used for the colony reduction test than for the turbidity reduction test, higher amounts of chimeric lysins were required in milk for significant antibacterial activities. Therefore, decreases in the antibacterial activities of chimeric lysins were apparent in milk.

### 2.8. Sequence Analyses of Chimeric Lysin Genes with Different Translation Efficiencies

Different translation efficiencies were apparent among the linker variants (>4-fold), SH3-domain variants (>10-fold), and different chimeric lysins (approximately 20-fold) (Figure 2, Figure 4, Figure 5 and Figure 6). The high- and low-expression proteins in *E. coli* had different amino acid compositions and synonymous codons, and the translation speed and the amounts of translated proteins are related to the context of the codons and the secondary structure of mRNA, as well as codon usages [48,49,50]. Additionally, the GC contents of the first 14–16 codons affect the translation efficiency of proteins [40]. Therefore, we analyzed the frequencies of amino acids and codons observed in high- and low-expression proteins, the presence of translation–stalling codon pairs, and the free energy of the secondary structure of partial and complete mRNA in all the tested chimeric lysins in this study (Appendix A). This is because the linker lengths increase the frequencies of high-expression-related amino acids, and the free energy of linkers and the complete mRNAs of linker variants decrease, and the expression levels tend to decrease, except for ALS-dA-L24. In the case of SH3-domain variants, the expression levels tend to increase with decreases in the frequencies of low-expression-related amino acids and codons and increases in the frequencies of high-expression amino acids. The relatively low expression level of ALS2-dA-L25 can be explained by the presence of translation–stalling codon pairs in the SH3 domain (PP and PG). The relatively high expression of lysostaphin can be explained by the higher free energy of complete mRNA than those of Lsp-ClyC_SH3_ and ClyC. However, the highest expression level of ClyO cannot be explained by any parameter. Even the first 15 codons of ClyO showed the highest GC content and lowest free energy (−10.0). We also compared the solubilities of the present chimeric lysins, and we could not find any correlations with expression levels [51].

As the peptidase domain was already codon-optimized, the ClyC_SH3_ domain was also codon-optimized to improve the expression level of Lsp-ClyC_SH3_. However, there was no effect. When we increased the cell-free expression temperature from 30 °C to 37 °C, the expression level (RI) decreased unexpectedly from 1.00 to 0.55. However, when we increased the incubation time to 20 h at 30 °C, the expression level (RI) increased from 1.00 to 1.33 (Appendix A). For Western blotting, we used soluble proteins after centrifugation, and we checked for the presence of insoluble chimeric lysins in the pellets using immunoblotting. However, we could not detect insoluble chimeric lysins. This result supports the theory that low-expressed chimeric lysin genes may themselves have lower translation efficiencies. In cases where the antibacterial activities are similar, more efficiently expressed EAD and CBD are preferred in the development of new chimeric lysins. Therefore, if a correlation between the expression levels of cell-free and *E. coli* expression systems is demonstrated, then the cell-free expression system may be useful for simply and rapidly verifying the real translational potency of predicted high-expression model genes. Additionally, different susceptibilities of different *S. aureus* strains were also, once more, observed, and the resistance mechanisms need to be unraveled [31].

## 3. Materials and Methods

### 3.1. Bacterial Strains

All *S. aureus* strains used in the experiments are listed in Table 1, and included 8 PMB strains, isolated from cased domestic bovine mastitis, previously reported in [43], 2 SNU strains isolated from chickens with arthritis or septicemia, previously reported in [31], and 10 CCARM strains previously reported from human cases [13,31]. *S. aureus* was cultured on tryptic soy agar (TSA) overnight at 37 °C incubation, and single colonies were inoculated into tryptic soy broth (TSB) and grown at 37 °C, with shaking at 200 rpm.

### 3.2. Bioinformatics

The SH3 domains of endolysin and phage lysin were predicted using the NCBI protein database (https://www.ncbi.nlm.nih.gov/protein, accessed on 24 February 2022). The alignment and sequence identity matrix of the SH3 domains were visualized and scored using the BioEdit program version 7.2.5 (https://software.informer.com/t/bioedit-windows-10/, accessed on 10 January 2023). The 3D structures of the natural linker protein were predicted using the Alphafold2 protein structure database (https://alphafold.ebi.ac.uk/, accessed on 3 March 2023), and the positions of amino acid residues were visualized using the UCSF Chimera program (https://www.cgl.ucsf.edu/chimera/, accessed on 3 March 2023) [52]. The relative intensity of the protein bands on the ECL Western blot was measured using the Image J software version 1.53 (http://imagej.nih.gov/ij, accessed on 15 September 2023). DNA secondary structure was predicted using vector builder (https://www.vectorbuilder.kr/tool/dna-secondary-structure.html, accessed on 2 January 2024).

### 3.3. Linker Design and Construction of Cell-Free Expression Genes

We previously reported the genomic sequence of ALS2 and its bactericidal activity against *S. aureus* [31]. We removed the amidase domain (116-321a.a) of ALS2 and used Alphafold2 to predict the structure of the naural linker. We bulit a total of 10 ALS2-dA-variants based on L38, which consists of 38 amino acids, by randomly removing 1–3 amino acid sequences.

To generate DNA templates for the cell-free expression system of each ALS2-dA variant, SH3 shuffling, and various chimeric lysins, we performed ligation via SOE-PCR [53]. An 87 base-pair segment of P_T7-RBS (NC_001604.1:22880-22966) was ligated to the front of the insert, and a 160 base-pair segment, containing the histidine, termination codon, and T7 terminator [32], was ligated to the back of the insert (Figure 1A). The primer sets for SOE-PCR and the amplification of ALS2-dA variants, SH3 shuffling, and various chimeric lysins are presented in Appendix A, respectively. The entire template was amplified via PCR, and amplicons were purified using the PureLink™ PCR Purification Kit (Thermo Fisher Scientific Inc., Waltham, MA, USA), according to the user manual. The final concentration of purified amplicons was 400–600 ng/1 μL, and purity was confirmed via 1.3% agarose gel electrophoresis. Cell-free protein synthesis was performed according to the manufacturer’s protocol for the Expressway™ Mini Cell-Free Expression System (Invitrogen). In brief, a final 50 μL reaction mixture consisting of 2.5× IVPS *E. coli* reaction buffer (-AA; 20 µL), *E. coli* slyD extract (20 µL), T7 enzyme mixture (1 µL), 50 mM amino acids (-met; 1.25 µL), 75 mM methionine (1 µL), DNA template (1 µg), and DNase-free distilled water was prepared. This mixture was incubated in a shaking incubator at 300 rpm for 30 min at 30 °C. Subsequently, a final 50 µL of 2× IVPS supply buffer (25 µL), 50 mM amino acids (-met; 1.25 µL), 75 mM methionine (1 µL), and DNase-free distilled water (22.75 µL) were mixed and incubated in a shaking incubator at 300 rpm for 5.5 h at 30 °C. After the completion of the reaction, the mixture was placed on ice for 5 min and then stored at 4 °C. For DNA-negative control samples, DNase-/RNase-free distilled water was used instead of DNA.

### 3.4. Western Blots Using Electrochemiluminescence for Detection of Cell-Free Synthesis Proteins

For the quantification of proteins synthesized using the cell-free synthesis system, the supernatant was obtained via centrifugation at 13,000× *g* for 10 min at 4 °C. Subsequently, 10 μL of the protein supernatant was mixed with 10 μL of RIPA Lysis and Extraction Buffer (Thermo Fisher Scientific Inc., Waltham, MA, USA), followed by the addition of 5 μL of 5× sample buffer with intermittent vortexing. The reaction was allowed to proceed for 2–3 h at room temperature. The reaction mixture was centrifuged at 13,000× *g* for 10 min at 4 °C, and 20 μL of the resulting supernatant was loaded onto Bis-Tris protein gels (4–12%, Invitrogen) for electrophoresis. Proteins were transferred to Immobilon-P membranes, and the membrane was blocked with 5% (*w*/*v*) skim milk powder in TBS (milk/TBS) for 1 h. After washing three times with TBS, the membranes were incubated overnight with HRP-conjugated goat anti-6-histidine polyclonal antibody (A190-113P; Bethyl Laboratories, Montgomery, TX, USA), diluted 1000-fold in 1% skimmed milk. The next day, the membrane was washed three times with TSB and developed using WesternBright™ ECL HRP substrate (Advansta., Menlo Park, CA, USA). Results were visualized using an ImageQuant™ LAS 4000 Mini-Biomolecular Imager (GE Healthcare, Pittsburgh, PA, USA) after adjusting for sensitivity.

### 3.5. Rapid Screening Testing of Cell-Free Synthesis Proteins

We have previously reported a rapid screening method to assess the antibacterial activity of cell-free proteins [31]. Briefly, single bacteria colonies were cultured overnight in TSB at 200 rpm in a 37 °C incubator. The grown bacteria were diluted to 2 × 10^7^ CFU/mL with TBS, and 100 μL was added to a 96-well plate. Varying amounts of protein, ranging from a maximum of 2 μL to a minimum of 0.06 μL, were mixed with the bacteria. The mixture was incubated at 37 °C with shaking at 200 rpm for 6 h, and the OD_600_ value was observed every hour.

### 3.6. E. coli Expression and Protein Purification

The genes encoding ALS2, ALS2-dA-L25, and ALS2-dA-ClyC_SH3_ were amplified using the primers listed in Appendix A and subsequently inserted into the Champion™ pET101 Directional TOPO™ Expression Kit (Thermo Fisher Scientific Inc.). The cloned genes were transformed into One Shot TOP10, according to the manufacturer’s protocol, and expressed in *E. coli* BL21(DE3). The protein induction was initiated using 0.1 mM isopropyl β-D-1-thiogalactopyranoside (IPTG) when the OD_600_ = 0.5 was reached, and the cultures were incubated at 20 °C for 20 h. The cultured cells were harvested via centrifugation at 3000× *g*, 4 °C, for 40 min, and the obtained pellet was resuspended in lysis buffer (50 mM Tris-HCl, 300 mM NaCl, 30% glycerol, pH 7.5). After sonication of the proteins using a sonicator (sonics-vcx750 model, Sonics and Materials Inc., Newtown, CT, USA), the proteins were mixed and purified using Ni-NTA resin. After washing three times with native wash buffer (50 mM NaH_2_PO_4_, 300 mM NaCl, 20 mM imidazole, pH 8.0), the proteins were eluted in several 1 mL columns using native elution buffer (50 mM NaH_2_PO_4_, 300 mM NaCl, 250 mM, or 500 mM imidazole). The eluted proteins were buffer-exchanged with protein storage buffer (50 mM Tris-HCl, 300 mM NaCl, 30% glycerol, pH 7.5) and filtered through a 0.45 μm pore-size filter. Protein content was quantified using the BCA assay, and the final proteins were analyzed via SDS-PAGE. SDS-PAGE was performed by loading purified proteins on Bis-Tris protein gels (4–12%, Invitrogen) and staining with Coomassie Brilliant Blue R-250 (Biosesang, Yongin-si, Republic of Korea).

### 3.7. Antibacterial Activity Testing of ALS2-dA-L25

The plate lysis assay was performed as described previously [27]. Briefly, a single colony of *S. aureus* was cultured in TSB at 37 °C 200 rpm overnight, and 1% of the cultured bacteria were inoculated in TSB and grown to mid-log phage (0.4~0.6). The grown bacteria were spread on TSA and air-dried for 10 min. The protein concentration was diluted to 10 μg, 1 μg, and 0.5 μg/10 μL using protein buffer. Next, 10 μL of the diluted protein was dropped on the plate and air-dried for an additional 10 min. After 20 h of incubation at 37 °C, the clear zone was observed the next day. For the turbidity assay, bacteria grown to mid-log phage (0.4~0.6) were centrifuged at 3000× *g* for 20 min according to an established method [54]. The obtained pellets were resuspended in protein buffer until OD_600_ = 1.0. Then, 200 μL of the suspension was dispensed into a 96-well plate, and 10 μg of protein was added to 10 μL. The mixture was shaken for 1 min, and the OD value was measured after 10 min.

The MIC was measured using a slight modification to the previous method [54]. *S. aureus* was grown to mid-log phage (0.4~0.6) at 37 °C, with shaking at 200 rpm. The bacteria were diluted to 10^5^ CFU/mL in TSB, and 50 μL was distributed to 96 wells. The purified protein was diluted 2-fold to 50 μL with protein buffer (50 mM Tris-HCl, 300 mM NaCl, 1 mM DTT (dithiothreitol), pH 7.5) in a 96-well plate. No growth was observed after 14~16 h of incubation at 37 °C.

### 3.8. Colony Reduction Testing in Milk

Fresh milk sourced from Seoul Milk (Seoul Dairy Cooperative, Seoul, Republic of Korea) was utilized in the experiment. The *S. aureus*, at a concentration of 10^3^ CFU/mL, was diluted in milk, and 100 μL of this dilution was added to each well in a 96-well plate. Then, 1 μL of cell-free synthesized protein was added to *S. aureus* in a 96-well plate and incubated at 37 °C for 20 min, with shaking at 200 rpm. The reaction was placed on ice and then plated on TSA overnight at 37 °C. The number of colonies was counted, and the colony reduction rate (%) of each chimeric lysin was calculated in comparison to the negative control.

### 3.9. Statistical Analysis

The statistical significance of the antibacterial activity was evaluated using the *t*-test and one-way analysis of variance (ANOVA), followed by Dunnett’s test in SPSS Statistics Windows, v. 26.0. All experiments were performed in triplicate for each sample. Triplicated and duplicated independent experiments were indicated.

## 4. Conclusions

The addition of the recently reported artificial T7 terminator and the reversion of mutated sequences to wild-type in the upstream of the coding region increased the expression levels of chimeric lysins. The improved cell-free expression system was successfully applied to select better chimeric lysins by adjusting different concentrations of the compared chimeric lysins. With the help of the improved cell-free expression system, amidase deletion, linker length optimization, and SH3 domain shuffling were verified to be effective approaches to improving the antibacterial activity of chimeric lysins, and a direct comparison of the antibacterial activities of contemporary chimeric lysins can be conducted rapidly in an easy manner. If the expression levels observed in the present cell-free expression system are correlated with *E. coli* expression levels, then it may also be useful to screen high-expression mutant genes. Additionally, the effects of the His-tag on the antibacterial activity of chimeric lysin need to be evaluated through further study.

## Figures and Tables

**Figure 1 ijms-25-02430-f001:**
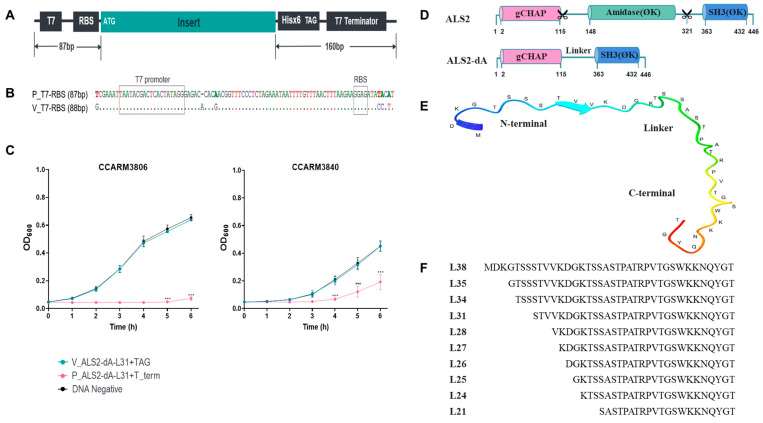
Comparison of the previous and the present cell-free expression systems and the generation of ALS2-derived amidase domain-deleted/linker-length variants. (**A**) The schematic structure of the gene template for improved cell-free expression and (**B**) the nucleotide difference between the previous (V_T7-RBS) and the present (P_T7-RBS) upstream regions from the coding gene. The different nucleotides identical to the natural nucleotide sequences of T7 promoter and RBS regions are shown. (**C**) Comparison of the antibacterial activities of ALS2-dA-L31 prepared using the previous and present cell-free expression systems against human MRSA strains (CCARM3806 and CCARM3840 (turbidity reduction test)). Data were analyzed using one-way ANOVA followed by Dunnett’s test to determine the significance relative to the negative sample without DNA template (*** *p* < 0.001). (**D**) The schematic structures of the reported ALS2 and the present amidase-deleted ALS2, ALS2-dA. (**E**) The predicted structure of the parent linker (L38) using Alphafold2 (blue: N-terminal, red: C-terminal). (**F**) The amino acid sequences of the tested linkers.

**Figure 2 ijms-25-02430-f002:**
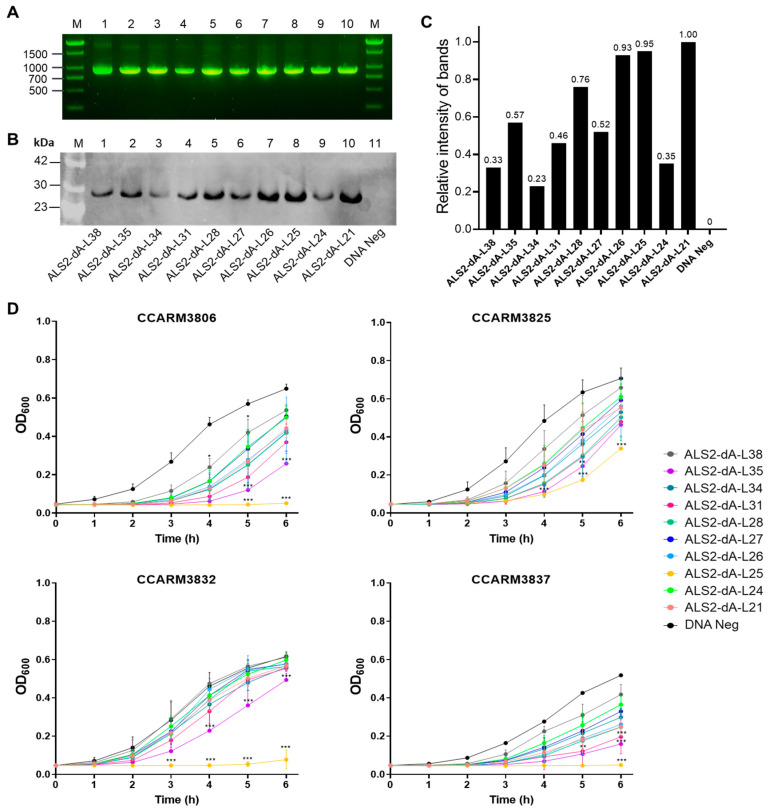
The DNA templates of the linker variants of ALS2-dA for cell-free expression and quantification and the antibacterial activity test of the expressed chimeric lysins. (**A**) High-concentration purified DNA for cell-free protein synthetic systems in ten linker variants. Samples at a concentration of 400–600 ng/μL were loaded with 2 μL each on 1.3% agarose gel. Lane M, 1000 bp size marker; Lane 1: ALS2-dA-L38 (527 ng), 967 bp; Lane 2: ALS2-dA-35 (524 ng), 958 bp; Lane 3: ALS2-dA-L34 (551 ng), 955 bp; Lane 4: ALS2-dA-L31 (490 ng), 946 bp; Lane 5: ALS2-dA-L28 (555 ng), 937 bp; Lane 6: ALS2-dA-L27 (539 ng), 934 bp; Lane 7: ALS2-dA-L26 (493 ng), 931 bp; Lane 8: ALS2-dA-L25 (547 ng), 928 bp; Lane 9: ALS2-dA-L24 (544 ng), 925 bp; Lane 10: ALS2-dA-L21 (550 ng), 916 bp. (**B**) Western blotting using ECL and (**C**) the relative protein intensity of the ten linker variants. (**D**) Comparison of the antibacterial activities of the ten linker variants. The antibacterial activities of chimeric lysins produced via a cell-free expression system were tested with a turbidity reduction test against human MRSA strains (CCARM3806, RST10-2; CCARM3825, RST2-1; CCARM3832, RST2-1; CCARM3837, RST4-1). Due to the different susceptibilities of the strains, different volumes of chimeric lysins were used: CCARM3806 and CCARM3837 (0.5 μL) and CCARM3825 and CCARM3832 (1 μL). The experiment was performed in triplicate and mean ± standard deviation (SD) values are shown. Data were analyzed using one-way ANOVA followed by Dunnett’s test to determine the significance relative to the negative sample without DNA template (*** *p* < 0.001, ** *p* < 0.01, * *p* < 0.05).

**Figure 3 ijms-25-02430-f003:**
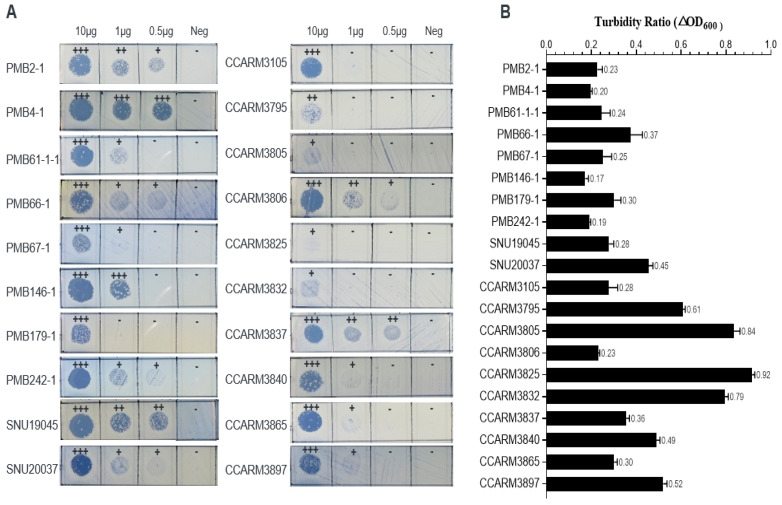
The antibacterial activity of *E. coli*-expressed ALS2-dA-L25. (**A**) Plate lysis test of the purified protein of ALS2-dA-L25. The protein formed a clear zone against various strains (isolated from bovine mastitis, chickens, and humans) at a concentration of 10/1/0.5 μg/10 μL. For negative samples, only protein purification buffer was used. (**B**) The turbidity reduction test for the antibacterial activity of the purified protein ALS2-dA-L25 against *S. aureus*. The turbidity ratio (TR) for each strain was determined by calculating the OD_600_ value ratio of the ALS2-dA-L25-treated to untreated samples.

**Figure 4 ijms-25-02430-f004:**
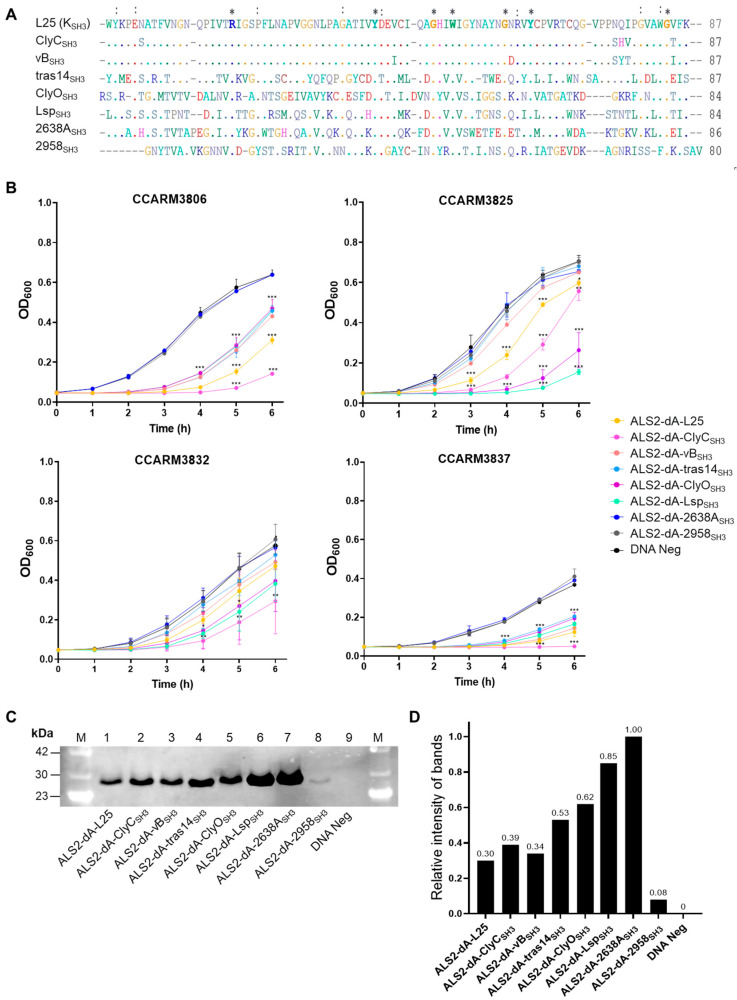
Comparison of the amino acid sequences of the SH3 domains and the antibacterial activities of SH3 domain chimeras in ALS2-dA-L25. (**A**) The amino acid lengths of the eight SH3 domains are presented as numbers. Amino acids common to all eight SH3s are marked with (*) and those common to seven are marked with (:). Amino acids common to seven sequences are marked with ALS2-dA-L25, and the respective SH3 domain-linked proteins were reacted with *S. aureus.* (**B**) Turbidity reduction tests for SH3 domain chimeras in ALS2-dA-L25 using CCARM3806, CCARM3825, CCARM3832, and CCARM3837 for 6 h. The experiments were performed in triplicate, and the mean and SD values are shown. Data were tested for significance using one-way ANOVA and Dunnett’s test compared to negative samples without DNA template (*** *p* < 0.001, ** *p* < 0.01, * *p* < 0.05). (**C**) Quantification of the SH3 domain chimeras of ALS2-dA-L25 using Western blotting. Lane M: protein molecular weight marker; Lane 1: ALS2-dA-L25; Lane 2: ALS2-dA-ClyC_SH3_; Lane 3: ALS2-dA-vB_SH3_; Lane 4: ALS2-dA-tras14_SH3_; Lane 5: ALS2-dA-ClyO_SH3_; Lane 6: ALS2-dA-Lsp; Lane 7: ALS2-dA-2638A_SH3_; Lane 8: ALS2-dA-2958_SH3_; Lane 9: DNA-negative sample. The molecular weight of the protein was approximately 26 kDa. (**D**) The relative intensity of each chimeric lysin was measured based on the relative intensity of each Western blotting band to ALS2-dA-2638A_SH3_ using Image J 1.53.

**Figure 5 ijms-25-02430-f005:**
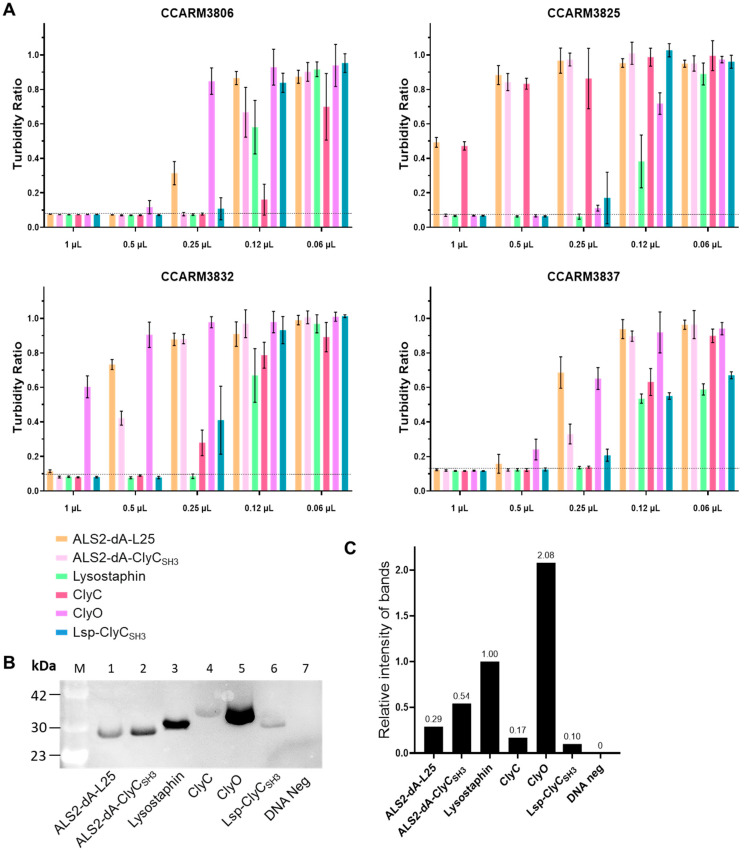
The determination of the minimal inhibitory volume of chimeric lysins (post-adjusting method). (**A**) Just after the cell-free expression of chimeric lysins, the antibacterial activity was measured using a turbidity reduction test against human MRSA strains (CCARM3806, CCARM3825, CCARM3832, and CCARM3837). The amount of protein used in the experiments ranged from 1 μL to 0.06 μL. The complete inhibition of bacterial growth was signified by the lack of changes in OD_600_ values during the 0–6 h incubation period, and the presence of dotted reference lines. The experiments were performed in triplicate, and the mean ± standard deviation (SD) values are shown. Below the dotted line, there was no growth of *S. aureus* during the 6 h of the reaction. (**B**) Western blotting of chimeric lysins using an anti-histidine antibody and ECL. Lane M: protein molecular weight marker; Lane 1: ALS2-dA-L25 (25.6 kDa); Lane 2: ALS2-dA-ClyC_SH3_ (25.6 kDa); Lane 3: Lysostaphin (28.1 kDa); Lane 4: ClyC (32.1 kDa); Lane 5: ClyO (29.6 kDa); Lane 6: Lsp-ClyC_SH3_ (28.1 kDa); Lane 7: DNA-negative sample. (**C**) The relative intensity of the protein bands. The ratio of protein expression compared to lysostaphin was measured using Image J 1.53.

**Figure 6 ijms-25-02430-f006:**
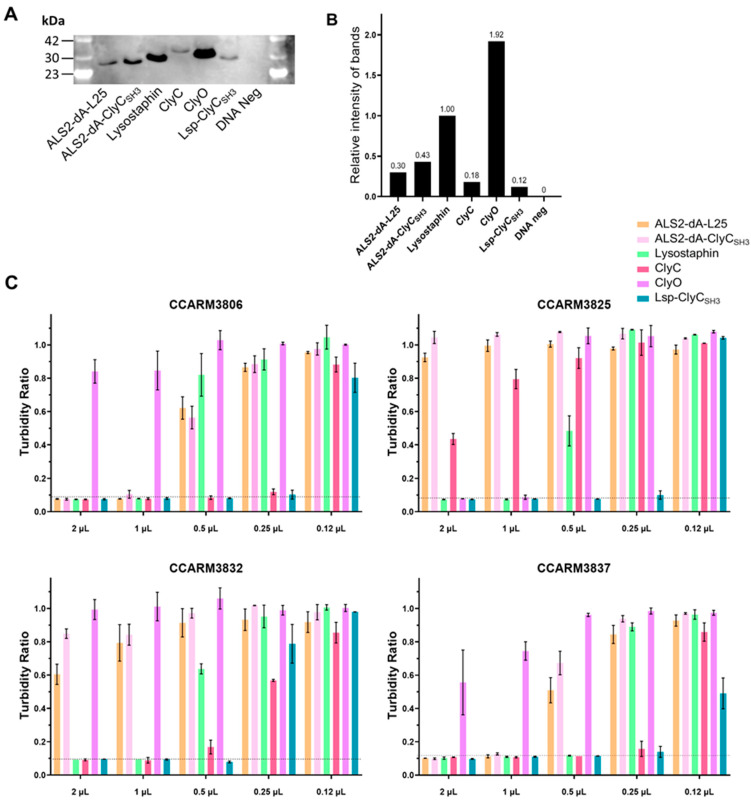
Direct comparison of the antibacterial activities of different chimeric lysins after adjusting the protein concentration to be equal (pre-adjusting method). (**A**) Western blotting of chimeric lysins using an anti-histidine antibody and ECL. Lane M: protein molecular weight marker; Lane 1: ALS2-dA-L25 (25.7 kDa); Lane 2: ALS2-dA-ClyCSH3 (25.7 kDa); Lane 3: Lysostaphin (28.1 kDa); Lane 4: ClyC (32.1 kDa); Lane 5: ClyO (29.6 kDa); Lane 6: Lsp-ClyCSH3 (28.1 kDa); Lane 7: DNA-negative sample. (**B**) The relative intensity of the protein bands. The ratio of the protein expression compared to lysostaphin was measured using Image J 1.53. (**C**) The amount of each chimeric lysin was adjusted based on the RI (0.12) of Lsp-ClyC_SH3_ to equal the final amount of protein: ALS2-dA-L25 (0.4 μL), ALS2-dA-ClyC_SH3_ (0.27 μL), lysostaphin (0.12 μL), ClyC (0.65 μL), ClyO (0.06 μL), and Lsp-Cly_SH3_ (1 μL). The proteins were reacted with the MRSA strains CCARM3806, CCARM3825, CCARM3832, and CCARM3837, and the turbidity ratio was calculated after 6 h. The amount of protein used in the experiments ranged from 2 μL to 0.12 μL. The complete inhibition of bacterial growth was determined by no changes in OD_600_ values during the 0–6 h incubation period, and the reference lines are dotted. The experiments were performed in duplicate, and the mean ± standard deviation (SD) values are shown. Below the dotted line, there was no growth of *S. aureus* during the 6 h of the reaction.

**Figure 7 ijms-25-02430-f007:**
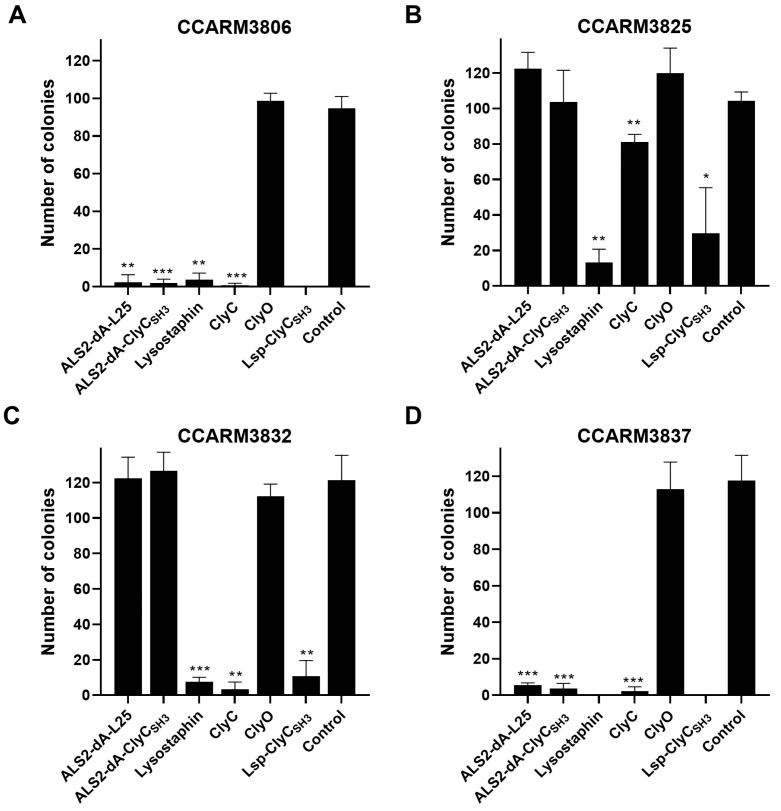
Comparison of the antibacterial activities of chimeric lysins in milk via a colony reduction test. According to the RI, the volume of each chimeric lysin was adjusted so that the final protein amount was the same: ALS2-dA-L25 (3.4 μL), ALS2-dA-ClyC_SH3_ (1.85 μL), lysostaphin (1 μL), ClyC (5.9 μL), ClyO (0.5 μL), Lsp-ClyC_SH3_ (10 μL), and the control (1 μL). The proteins were reacted with 10^3^ CFU/mL of *S. aureus—*(**A**) CCARM3806, (**B**) CCARM3825, (**C**) CCARM3832, and (**D**) CCARM3837—in ultra-high-temperature processed commercial milk for 20 min. Experiments were performed in triplicate and mean ± standard deviation (SD) values are shown. Data were subjected to a *t*-test to determine the significance compared to the control (*** *p* < 0.001, ** *p* < 0.01, * *p* < 0.05).

**Table 1 ijms-25-02430-t001:** *S. aureus* strains used in this study and their susceptibility to ALS2-dA-L25.

Strain	Origin ^1^	RST ^2^	MRSA	Strain	Origin ^1^	RST ^2^	MRSA
PMB2-1	BM	10-2	-	CCARM33105	HI	4-1	+
PMB4-1	BM	22-1	-	CCARM3795	HI	4-1	+
PMB61-1-1	BM	10-2	-	CCARM3805	HI	2-1	+
PMB66-1	BM	14-3	-	CCARM3806	HI	10-2	+
PMB67-1	BM	5-2	-	CCARM3825	HI	2-1	+
PMB146-1	BM	11-5	-	CCARM3832	HI	2-1	+
PMB179-1	BM	4-1	-	CCARM3837	HI	4-1	+
PMB242-1	BM	11-1	-	CCARM3840	HI	4-1	+
SNU19045	CA	3-1	-	CCARM3865	HI	2-1	+
SNU20037	CA	3-1	-	CCARM3897	HI	4-1	+

^1^ BM, bovine mastitis; CA, chicken arthritis; HI, human infection. ^2^ RST: *rpoB* sequence type [44].

**Table 2 ijms-25-02430-t002:** The minimum inhibitory concentration (MIC) of the expressed protein.

	MIC (µg/mL ^a^)			
Strain	ALS2	ALS2-dA-L25	ALS2-dA-ClyC_SH3_	Lysostaphin
	(MW = 51.5 KDa)	(MW = 27.3 KDa)	(MW = 27.2 KDa)	(MW = 26.9 KDa)
	Purity = 72.4%	Purity = 91.1%	Purity = 80.2%	Purity = 90.8%
PMB4-1	22.5 ± 9.1 (430 nM)	4.7 ± 1.5 (170 nM)	2.5 (90 nM)	0.15 (5 nM)
SNU19045	36.2 (700 nM)	11.4 (420 nM)	5.2 (190 nM)	0.15 (5 nM)
CCARM3806	54.3 (1050 nM)	22.8 (830 nM)	15.0 (550 nM)	0.15 (5 nM)

^a^ The MIC was adjusted to the purity of each chimeric lysin.

**Table 3 ijms-25-02430-t003:** Adjusted and experimental minimal inhibitory volumes (µL) of chimeric lysin.

Strains	ALS2-dA-L25	ALS2-dA-ClyC_SH3_	Lysostaphin	ClyC ^1^	ClyO ^2^	Lsp-ClyC_SH3_
CCARM3806	0.15 (0.50)	0.14 (0.25)	0.25 (0.25)	0.04 (0.25)	2.08 (1)	0.05 (0.50)
CCARM3825	>0.29 (>1)	0.54 (1.00)	0.25 (0.25)	>0.17 (>1.00)	1.04 (0.5)	0.05 (0.50)
CCARM3832	>0.29 (>1)	0.54 (1.00)	0.25 (0.25)	0.09 (0.50)	>2.08 (>1.00)	0.05 (0.50)
CCARM3837	0.29 (1.00)	0.27 (0.50)	0.25 (0.25)	0.04 (0.25)	2.08 (1.00)	0.05 (0.50)

^1^ References [24] and ^2^ [33].

## Data Availability

Data are contained within the article or the Appendix A.

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
