# Peer review of "Rapid Antibacterial Activity Assessment of Chimeric Lysins"

_ijms, 2024, doi:10.3390/ijms25042430_

Round 1
Reviewer 1 Report
Comments and Suggestions for Authors
Dear Authors
I have reviewed your manuscript Rapid Antibacterial Activity Assessment of Chimeric Lysin and find it very interesting. I have few comments as below:
1. Abstract and introduction are well presented.
2. Please add ClyV endolysin (10.1007/s00253-019-10325-z) in the introduction as one of the potent chimeric lysin.
3. Please improve fig 1B, unclear.
4. Comparison of the antibacterial activities of chimeric lysins in milk via a colony reduction test should be depicted in Log10CFU/ml, not in %age reduction.
5. How was the purity of these cell lysates was determined? SDS PAGE of protein will be helpful in determining the purity of protein over western blots only.
Thank you.
Comments on the Quality of English LanguageIn general the paper is well written and clear to scientific readers.
Reviewer 2 Report
Comments and Suggestions for Authors
In the MS ijms-2862092, the authors are focused on a priority field: finding new antibacterial agents effective against MDR bacteria S. aureus. They continue previous research on chimeric lysins as follows:
- They described a method to increase the chimeric lysins' expression levels. Thus, the expressed protein amounts were compared, and the antibacterial activities of different chimeric lysins were evaluated, establishing the most active ones;
- They modified ALS2 by deleting the amidase domain, optimizing the linker between the CHAP and SH3 domains, and shuffling with various SH3 domains. Then, the higher antibacterial potential of the newly synthesized chimere lysins was demonstrated.
The Introduction offers enough data to justify the emergence of such research. The results are presented point by point, with suggestive figures and graphs, well-performed.
The conclusions are reasonable and supported by discussion and statistical significance determination.
Minor comments and suggestions are available below:
1. Maybe using "Lysins" in the title could offer more clarity to the reader than "Lysin" because the present study evaluates more than one.
2. Due to the study's complexity, the authors are encouraged to verify the English editing again regarding the specific terms and phrases formulation.
Reviewer 3 Report
Comments and Suggestions for Authors
Jin-Mi and co-authors improved the cell-free expression system to boost the yield of chimeric lysins that are used for antibacterial applications. This work included multiple experimental approaches and some of the results looked promising. I would recommend this manuscript to be published if some minor questions could be answered:
In figure 1C, how different is the previous method (V_ALS2-dA-L31+TAG) from the negative control? Based on the results, it seems that they are highly similar. Does it mean that the previous method does not work at all?
Does figure 2C come from only one experiment (2B)? Could the authors repeat the assay at least two times more to show statistical significance of the difference in protein expressions?
Page 11 line 370-371 mentions that adding His-tag may decrease the antibacterial activity. Is there any reference? Could the authors elaborate on this?
Section 2.8 reads confusing and I personally don’t see the necessity of discussing about the sequence analysis. If the authors insist to include this section, please also include repeat experiments for the expression test in Fig S2.
Some typos:
Page 2, line 50, phages
Page 2, line 68, which is
Page 2, line 78, the same
Page 11, line 345, after confirming the stability
Page 11, line 355-356, stronger antibacterial activities and broader antibacterial spectrum
